# Widespread but Poorly Understood Bacteria: Candidate Phyla Radiation

**DOI:** 10.3390/microorganisms10112232

**Published:** 2022-11-11

**Authors:** Yanhan Ji, Ping Zhang, Sihan Zhou, Ping Gao, Baozhan Wang, Jiandong Jiang

**Affiliations:** Department of Microbiology, College of Life Sciences, Nanjing Agricultural University, Key Laboratory of Agricultural and Environmental Microbiology, Ministry of Agriculture and Rural Affairs, Nanjing 210095, China

**Keywords:** candidate phyla radiation (CPR), patescibacteria, metabolic characteristic, ecological distribution, episymbiosis

## Abstract

Candidate Phyla Radiation (CPR) bacteria is a bacterial division composed mainly of candidate phyla bacteria with ultra-small cell sizes, streamlined genomes, and limited metabolic capacity, which are generally considered to survive in a parasitic or symbiotic manner. Despite their wide distribution and rich diversity, CPR bacteria have received little attention until recent years, and are therefore poorly understood. This review systematically summarizes the history of CPR research, the parasitic/symbiotic lifestyle, and the ecological distribution and unique metabolic features of CPR bacteria, hoping to provide guidance for future ecological and physiological research on CPR bacteria.

## 1. Research History of CPR Bacteria

Candidate phyla radiation (CPR), classified under the bacteria domain, is the radiation of major lineages defined as candidate phyla due to their lack of isolated representatives [1]. These microorganisms are ultra-small, have a streamlined genome, and possess unusual ribosomal compositions, and part of these microorganisms contain intron sequences within 16S rRNA genes, which is a feature rarely reported in bacteria [1]. The first sequence of the CPR lineage was a 350 bp fragment of a 16S rRNA gene clone named TM7 (abbreviation of Torf Mittlere schicht clone 7), from an environmental sample (Figure 1) [2]. In order to detect the bacterial diversity in a peat sample from Germany, 16S rRNA genes of total environment genomic DNA were determined by clone library consultation and hybridization with oligonucleotide probes, and a clone named TM7, which shared less than 75% similarities with all known phyla, was found to form a relative deep monophyletic branch among the bacteria domain. With the development and application of next generation sequencing technology, more and more candidate phylum bacteria have been detected in diverse habitats [3,4,5,6,7]. Patescibacteria, the superphylum accommodating the three candidate phyla OP11, OD1 and GN02, reproducibly formed a monophyletic group based on concatenated marker gene analysis in the investigation of uncultured microorganisms from different habitats by single-cell genomics [8]. The formation of monophyletic branches on the evolutionary tree by candidate phyla is not limited to these three phyla. Combined with size fraction and metagenomic sequencing, 797 draft-quality bacterial metagenome-assembled genomes (MAGs) belonging to 35 phyla were obtained from groundwater in Rifle, CO, USA [1]. These CPR microorganisms have ultra-small cells, limited metabolic capacity, and unusual ribosome compositions, and there are self-splicing introns and protein insertions in the 16S rRNA gene, which is rarely reported in normal bacteria [1]. The phylogenetic analysis shows that CPR is monophyletic, based on the 16S rRNA gene tree and ribosomal concatenated protein tree (Figure 1). These unique subminiature bacteria with a shared evolutionary history were described as candidate phyla radiation, which includes the superphylum “Patescibacteria” [1]. As of 2018, CPR has been extended to 70+ phylum-level lineages, but rank normalization according to relative evolutionary distance suggests that CPR should be amalgamated into one phylum, and the name Patescibacteria was thus used [9]. According to the criteria of a 75% threshold of bacterial 16S rRNA gene sequence at the phylum level, CPR may account for >15% of bacterial diversity [1], but other studies also suggest that CPR may account for more than 25% of bacterial diversity [10,11]. In general, CPR accounts for a large fraction of bacterial diversity, yet much of what we know about CPR remains at the genomic level. With the development and application of sequencing techniques, a growing number of CPR taxa have been detected, but only a few strains have been isolated in co-culture with their hosts [12,13,14,15].

## 2. The Parasitic or Symbiotic Lifestyle of CPR Bacteria

### 2.1. Unique Genomic and Morphological Characteristics

Small cell sizes and streamlined genomes are the most common features of CPR bacteria. Combining metagenomic analysis with size fraction (1.2–0.2–0.1 μm), CPR bacteria were found to be enriched on small pore size filters (0.1/0.2 μm) [1,16]. Cryogenic transmission electron microscopy results also demonstrated that CPR bacteria have an ultra-small cell size, and some CPR bacteria can pass through a 0.2 μm filter [17]. In 2015, He et al. isolated and co-cultured a CPR strain TM7x, with its bacterial host (*Actinomyces* spp. XH001) from the human oral cavity. The CPR strain TM7x was spherical with a diameter of 200–300 nm, as judged by microscopic observation [12]. In 2021, Batinovic et al. isolated another CPR strain JR1 from wastewater, which could violently lyse foaming strains (*Gordonia amarae*) [13]. The cell size of strain JR1 observed by electron microscopy was between 0.2 μm and 0.5 μm. Through electron microscope observation, the morphology of CPR bacteria is determined to be mainly rod or spherical.

The genome sizes of all CPR bacteria obtained so far are less than 1.5 Mbp. Compared with normal bacteria, archaea and obligate insect symbionts, the genome size of CPR bacteria is closer to that of obligate insect symbionts, and significantly smaller than that of most normal bacteria [18]. Castelle et al. analyzed and compared approximately 1000 CPR genomes, and found that most of the CPR genomes had genes related to homologous recombination, base excision repair and mismatch repair, suggesting that the streamlined genome of CPR bacteria is a trait inherited from their ancestors, rather than a reduction in genome evolution [18]. After comparing three genomes of CPR bacteria of Absconditabacteria (SR1), Moreira et al. found that there were only 390 conserved genes in this type of CPR bacteria [19]. Based on the comparative study of CPR genomes among three different taxa (SR1, GN02, PER), it was found that these three CPR phyla may have recently lost 30–50% of genes from their common ancestor, and it was speculated that the genome content of these CPR bacteria had an active dynamic evolution [19]. Whether the streamlined genome of CPR is just an ancestral trait or the result of reductive evolution still requires further investigation.

Interestingly, CPR bacteria often have self-splicing introns and proteins encoded within their rRNA genes. In total, 1543 CPR bacterial 16S rRNA genes (>800 bp) assembled from the groundwater near the town of Rifle in USA were clustered into 713 sequences at 97% identity. It was found that 31% of 16S rRNA genes encoded insertion sequences larger than 10 bp, and most of the sequences over 500 bp encoded a catalytic RNA intron or an open reading frame (ORF), suggesting that CPR bacteria were self-splicing, which was supported by metatranscriptomic analysis (Figure 2) [1]. Based on the analysis of about 1000 CPR genomes, it was found that the phenomenon of an insertion sequence also existed in 23S rRNA and tRNAs genes [18]. A study on primer fidelity in SSU (small subunit ribosome) rRNA gene sequences suggested that 70% of the SSU rRNA gene sequences of CPR bacteria would be missed when using bacterial universal primer 515F-806R [20]. Furthermore, the ribosomal protein composition of CPR bacteria is also different from that of other bacteria. CPR bacteria usually lack some ribosomal proteins that are ubiquitous in other bacteria, such as ribosomal proteins uL1, bL9, and uL30, and such features are similar to some parasitic microorganisms [1]. Further, the ribosomal proteins bL28, uL29, bL32, and bL33 are also lost in some specific lineages of CPR bacteria [21]. In addition, several regions in ribosomal proteins as well as in the 16S, 23S, 5S rRNAs of CPR bacteria were lacking, and these missing regions were predicated to be located near the surface of the ribosome, implying that CPR bacteria might possess smaller ribosomes with more simplified surface structures than other non-CPR bacteria [21].

Together, the virus-sized cell sizes, self-splicing introns and proteins encoded within rRNA genes, and mismatch between the universal bacterial 16S rRNA gene primers and SSU rRNA gene sequences of CPR make the detection of CPR bacteria in environments especially difficult.

### 2.2. Potential Metabolism

The streamlined genomes limit the metabolic capacity of CPR bacteria. Current studies have shown that most CPR bacteria lack a complete respiratory chain, including the NADH dehydrogenase and oxidative phosphorylation complex, and the tricarboxylic acid (TCA) cycle, suggesting that CPR bacteria may be anaerobic (Figure 2) [16]. CPR bacteria have an incomplete glycolytic pathway, and lack 6-phosphofructokinase and glucokinase. CPR bacteria usually contain genes related to the Pentose Phosphate Pathway (PPP). Therefore, it is speculated that CPR bacteria can convert fructose-6-phosphate to glyceraldehyde-3-phosphate through the non-oxidative PPP pathway [18]. Usually, pyruvate or acetyl-CoA is the end-product of central carbon metabolism in CPR bacteria. They generally first convert pyruvate to acetyl-CoA, and then further use it to produce short-chain fatty acids. For example, many CPR bacteria such as parcubacteria (OD1), dojkabacteria (WS6) and microgenomates (OP11) are predicted to utilize the ADP-acetyl-CoA synthase (ADP-Acs) commonly found in archaea to generate acetate [23]. Some peregrinibacteria (PER) can also produce acetate via acetate kinase (Ack) and phosphotransacetylase (Pta), which are common in bacteria. In addition to acetate, many CPR bacteria can produce products such as lactic acid, formic acid, or ethanol through fermentation [18].

The lack of complete amino acid, nucleotide and lipid synthesis pathways is a common feature of CPR bacteria, but the biosynthetic capacities of different lineages of CPR bacteria vary greatly [18]. Certain genomes from peregrinibacteria (PER) have relatively more core biological metabolic capabilities, and all seem to have the ability to synthesize nucleotides, certain amino acids, and cofactors, but not fatty acids [24]. Compared to peregrinibacteria (PER), katanobacteria (SM2F11), KAZAN and WS6 have the least biosynthetic and metabolic capabilities, and they lack the ability to synthesize nucleotides, amino acids, lipids, peptidoglycan, and various auxiliaries [18]. The great metabolic differences among CPR bacteria suggest that although all CPR bacteria are predicted to require parasitism or symbiosis to survive, different lineages of CPR may differ in degrees of dependence on their host.

### 2.3. Isolation and Culture of CPR

With the wide application of high-throughput sequencing technology, an increasing number of CPR bacteria have been detected in various environments, but most studies are mainly based on the metagenome-assembled genomes (MAGs), lacking the isolates of a pure culture of CPR [1,25,26]. Among the more than 70 phylum-level lineages of CPR bacteria, only a few lineages have been isolated in binary cultures that could be stably passaged, such as saccharibacteria [12,14], which are summarized in Table 1. Furthermore, although most CPR bacteria are inferred to be symbionts, there is still a lack of reliable and standardized methods to obtain pure binary cultures of CPR bacteria and their hosts.

#### 2.3.1. Saccharibacteria (TM7)

The first pure strain of CPR bacteria, TM7x, was isolated from a human oral cavity [12]. By serial sub-cultivation with the addition of streptomycin (given the resistance of TM7x to streptomycin), an enrichment-containing strain TM7x and multispecies was obtained, then the pure binary co-culture of strain TM7x with its host, *Actinomyces odontolyticus* strain XH001, was isolated using SHI medium (a medium for culturing saliva-derived oral microbial) agar plate. The growth of TM7x was obligately dependent on its host strain XH001, and neither the addition of spent co-culture medium nor heat-killed XH001 could lead to the independent growth of TM7x [12]. Further studies reveal that strains TM7x/XH001 could co-exist well under nutrient-replete condition, but the infection of strain XH100 with strain TM7x would lead to a severely disrupted cell membrane in the host and a consequently decreased viability of strain XH001 under extended starvation conditions, suggesting a parasitic rather mutualistic or commensal relationship between strains TM7x and XH001 [12].

Strain TM7x elicited differential responses to its different hosts. For some hosts, the rapid growth of strain TM7x at the initial stage of co-cultivation would lead to a large number of deaths of its hosts [28]. After several passages, hosts would rapidly evolve to adapt to strain TM7x and form a long-term stable binary culture. Due to the genetic changes in the host during several passages, the growth collapse phenomenon of the host only occurred at the early stage of infection [28]. However, not all related *Actinomyces* strains showed such high susceptibility to strain TM7x, and some exhibited no growth–crash phase when infected by strain TM7x before establishing a stable binary co-culture [29]. The further comparative genomic analysis revealed only very small genetic differences between hosts with varying susceptibility to strain TM7x [29]. Furthermore, the acquisition of the arginine deiminase system (ADS) could allow strain TM7x to maintain higher activity and infectivity when disassociated from its host, and protected strain TM7x and its host from acid stress in oral cavity [30]. It was intriguing that strain TM7x containing ADS only preferred the hosts lacking the ADS system, and not those carrying ADS, suggesting the importance of ADS to the partner selection for episymbiosis within the mammalian microbiome [30].

Based on 16S rRNA gene sequencing, it was found that the abundance of TM7 in the mouth of patients with periodontitis was higher than that of healthy people, indicating that TM7 might be a pathogen. However, through the mouse oral infection model, it was found that strain TM7x can provide protection to the mammalian host by reducing the pathogenicity to the host, indicating that strains with increased abundance in disease were not necessarily harmful [31]. In addition, is TM7 also associated with the phenomenon of preying on hosts, lysing hosts like a virus. CPR strain (TM7-JR1) was isolated from a wastewater by an accidental attempt [13]. *Gordonia amarae*, one of the most common foaming bacteria, has multiple antiviral mechanisms within its genome. In order to isolate phages that could lyse G. *amarae*, the in situ wastewater was filtered through a 0.45 μm filter membrane, and then the filtrate was directly applied to a colony of G. *amarae*, but one CPR strain (TM7-JR1) that could strongly lyse G. *amarae* was accidentally isolated. The lifestyle of strain TM7-JR1 that lyses its host like a virus provides a new perspective on the relationships between CPR and its host.

#### 2.3.2. Absconditabacteria (SR1)

The first in-depth characterization of Absconditabacteria (SR1) was based on the environmental observation and genome-inferred biology of a strain named “*Candidatus Vampirococcus lugosii*” [19]. Many photosynthetic bacteria were observed with one or several enigmatic small and dark nonflagellated cells attached to their surface in an enrichment cultured from microbial mats collected from a permanent hypersaline lake in Western Europe; in some cases, the epibionts were associated to empty ghost cells where only the photosynthesis-derived sulfur granules persisted [19]. These epibionts were called “*Candidatus Vampirococcus lugosii*” because the phenotypic characteristics were in agreement with the genus *Vampirococcus* observed over forty years ago, and might belong to the same genus. 16S rRNA gene sequencing and analysis of both the epibiont and the host for ten infected cells collected by the micromanipulator found that the two bacteria were halochromatium-like γ-proteobacterium and SR1. Combined with micromanipulation and whole-genome amplification sequencing, two MAGs were obtained, belonging to the gamma-proteobacteria photosynthetic bacteria and the candidate phylum SR1 of CPR bacteria, respectively, in which the genome of strain “*Candidatus Vampirococcus lugosii*” was relatively complete, while the host genome was partially assembled (15%), indicating that the epibiont might consume the host DNA [19]. Besides this, the genome sequence of SR1 obtained in this study coded a very simplified metabolism and a complex cell surface, implying that the attachment of SR1 to its host by type IV pili is needed to obtain all cell components it needs [19]. The first stably cultivated species of SR1, “*Ca. Absconditicoccus praedator*” M39-6, was isolated from the hypersaline alkaline Lake Hotontyn Nur, Mongolia. M39-6 appeared to share a similar biology with the other genus “*Candidatus Vampirococcus lugosii*”: an obligate parasitic lifestyle, feeding on photosynthetic anoxygenic γ-proteobacteria, and the complete consumption of the host cytoplasm, suggesting that predation on hosts might be a common feature of microorganisms in this lineage [14].

Cross-domain parasitism between CPR bacteria and their hosts has also been observed. A strain of CPR bacteria (yanofskybacteria), specifically parasitizing methanogenic archaea *Methanothrix*, was isolated from wastewater treatment sludge samples [15]. The cellular deformation and reduced activity of *Methanothrix* filaments (multicellular) attached to Ca. yanofskybacteria implied that the interaction was parasitic [15]. Transmission electron microscopy and FISH with specific probes showed that protists might be important hosts for some CPR bacteria [27]. The host range of CPR bacteria includes more than just bacteria, but the study of the relationship between CPR bacteria and their hosts is still limited by the lack of isolates.

## 3. Ecological Distribution and Function of CPR Bacteria

CPR bacteria are rich in species diversity, predicted to account for 25% or more of bacterial diversity. With the development and application of sequencing technology, CPR bacteria have expanded to 74 phylum-level lineages from the initial 35 lineages [18]. However, due to their small cell size, the insertion sequence in the3 16S rRNA gene, and the preference of universal primers, the species diversity of CPR bacteria is far from fully explored.

### 3.1. Extensive Ecological Distribution of CPR Bacteria

In recent years, the distribution of CPR bacteria in natural environments has been widely reported. At present, the research on CPR bacteria is predominantly based on the water environment, including freshwater lakes [32], seawater [33,34,35], wastewater treatment plants [36,37], groundwater [1,17] and other aquatic ecosystems. Especially in groundwater, CPR bacteria accounted for 20% or more of the total bacterial community [26]. In addition, CPR bacteria have also been reported to be distributed in sediments [38], plant rhizosphere [39], and soils [40,41]. Interestingly, the presence of CPR bacteria has also been detected in animals [42] and humans [43] and is thought to be potentially relevant to human health. It can be seen that CPR bacteria are widely distributed in different ecosystems and might play an important role in many biogeochemical cycles. However, the distribution of CPR bacteria in different habitats and whether CPR bacteria have habitat preferences are still unclear.

Metagenomic analysis of one agricultural and seven pristine groundwater samples found that CPR bacteria accounted for 3–40% of the groundwater bacterial community [26]. The similarity of bacterial communities in groundwater was evaluated at the genome level, and it was found that there was a lack of similarity at the levels of phylum, species and strain in different groundwaters, indicating that CPR bacteria might be differentiated according to physicochemical conditions and host populations. Combining cryogenic transmission electron microscopy imaging with genomic analyses, it could be hypothesized that the growth of CPR bacteria was stimulated by attachment to the host surface [26]. The analysis of 119 metagenomic samples from 19 freshwater lakes in Eurasia showed that the CPR community in freshwater lakes was dominated by OP11 and OD1, while PER and TM7 accounted for a small proportion of bacterial communities [25]. The distribution characteristics of CPR bacteria in different lineages was independent of nutrient conditions, and similar MAGs (ANI > 98%) were obtained in distant sampling sites, suggesting that the distribution of CPR bacteria may be related to their hosts. For freshwater lake samples, some CPR bacteria were observed to be attached to other bacterial surfaces in the environment, or parasitized in protozoa by CARD-FISH labeling, and a part of the CPR bacteria were also observed to float freely in the environment or adhere to extracellular aggregates. It was speculated by the analysis of genome replication rate that CPR bacteria may need to attach to the host to enact their replication activity [25]. Our understanding of the ecological distribution of CPR bacteria remains limited, which means that more systematic studies are needed to reveal the distribution and species diversity of CPR bacteria in different environments.

### 3.2. Participation of CPR Bacteria in Biogeochemical Cycle

Although all CPR bacteria share a fermentative-based lifestyle, CPR bacteria are metabolically diverse and may play an important and critical role in biogeochemical circulation.

CPR bacteria participate in the carbon cycle and have the potential to assimilate carbon dioxide (CO_2_) and degrade complex carbon substrates. Combining flow cytometry sorting with metagenomics, the genome of SR1 was reconstructed from human oral samples. A gene encoding an archaeal-type 1,5-diphosphate ribulose carboxylase (Rubisco) was found in the genome, with the potential to fix carbon dioxide, as found in the PER phylum [16,44,45]. A study has shown that the Rubisco gene of CPR bacteria could define a monophyletic branch, similar to the archaeal form III Rubisco [16]. The PER form II/III Rubisco expressed in a photoautotrophic Rubisco-deleted strain was able to fix CO_2_ to supplement phototrophic growth, indicating its catalytic activity [45]. Furthermore, genomic information from different habitats indicated that CPR bacteria have the metabolic potential to degrade complex carbon substrates. A total of 135 different glycoside hydrolase (GH) families were found in over 2000 CPR bacterial genomes [46]. In all CPR bacteria, although the ability to degrade substrates (such as amylose and cellulose) is similar, the genes encoding glycoside hydrolases are different. The CPR bacterial genomes constructed from thermokarst lakes possessed an average of 29 ± 12 carbohydrate-active enzyme (CAZy) genes identified per genome, and those MAGs received from freshwater lakes in Eurasia contained an average of 15 CAZy genes per genome, indicating that the ability to degrade both complex and simple carbon substrates is common in CPR bacteria, and may differ among CPR bacteria from different environments [25,47].

Moreover, genes involved in denitrification have also been found in the genomes of many CPR bacteria, suggesting that CPR bacteria may also be involved in the nitrogen cycle. The nitrogen cycle is one of the basic material cycles in the biosphere, and denitrification is an important part of the nitrogen cycle. Many studies have shown that CPR bacteria may participate in the nitrogen cycle in the geochemical cycle [16,46]. In total, 71 CPR bacterial genomes were reconstructed from groundwater samples by metagenomic analysis, including four independent MAGs from two different lineages (Kaisercharacter and Harrisonbacteria), which contained nitrite reductase genes (*nir*K, the key gene in the process of denitrification) [46]. This copper containing nitrite reductase can reduce nitrite to nitric oxide. At the same time, the obtained *nir*K gene sequence of CPR bacteria was compared with other *nir*K gene sequences and a phylogenetic tree was constructed. It was found that the *nir*K gene of CPR bacteria could form a separate evolutionary branch on the evolutionary tree. In addition, studies have found that a special individual “*Candidatus Parcunitrobacter nitroensis*” in CPR bacteria had a basically complete electron transfer chain, and its genome encodes all enzyme genes involved in the nitrogen respiration process (nitrate reductase, hydroxylamine redox enzyme and nitric oxide reductase) [48]. Similarly, their sequences were significantly different from those of isoenzymes in other organisms. More and more studies have shown that other CPR bacterial genomes also contain nitrate reduction-related genes [49].

Sulfur is one of the essential macronutrients for organisms, and microorganisms play an important role in the sulfur cycle. CPR bacteria may also be involved in sulfur redox. In total, 49 MAGs were obtained by sequencing groundwater samples in Colorado, and it was found that the genomes of two lineages of CPR bacteria, OD1 and OP11, contained 3 Fe-hydrogenase- (Fe-hydrogenase) and 23 NiFe-hydrogenase (NiFe-hydrogenase)-encoding genes. Further phylogenetic analysis found that 17 nickel–iron hydrogenases were highly similar to the *3b* cytoplasmic hydrogenases of the archaea Thermococcales, suggesting that they may be involved in the reduction of polysulfides [16]. In the CPR bacterial genome reconstructed from groundwater in California, a sulfur dioxygenase gene (*sdo*) was found, as well as many sulfate reduction-related genes (*sat*, *cys*C and *cys*N), suggesting that CPR bacteria may play a role in the sulfur cycle [26].

## 4. Perspective

The rapid development of omics technology in recent years has provided a powerful tool to study the ecological distributions, potential physiological and metabolic characteristics, and genetic evolutions of uncultured microorganisms, such as CPR bacteria, in natural and anthropogenic environments. Considering the high diversity, ubiquitous distribution and unique biological characteristics of CPR bacteria, at least the following research directions of CPR bacteria should be pursued in the future:

(1) Species diversity and ecological distribution—CPR bacteria may account for about one-quarter of the bacterial diversity of the planet, but currently the vast majority of the genomes of CPR bacteria are obtained from groundwater and a small amount from wastewater treatment systems and other habitats, which consequently leads to an urgent need to study the diversity and ecological distribution patterns of the CPR bacteria of ecological importance in other natural and anthropogenic environments;

(2) Pure culture and physiological characteristics—Before now, only a few pure cultures of CPR bacteria have been co-isolated with their hosts, which severely obstructs our understanding of the physiological and metabolic characteristics of CPR bacteria, as well as the mechanisms underlying their symbiotic lifestyle with their hosts. Thus it is crucial to develop technologies to isolate pure binary cultivations and CPR strains for biological feature studies;

(3) Ecological functions and ecosystem importance—Genomic analysis reveals plenty of functional genes involved in the biogeochemical cycle of carbon, nitrogen, sulfur, etc., in many lineages of CPR bacteria. Moreover, the epiparasitic features of most CPR bacteria inevitably directly or indirectly influence microbial community structures and functions via affecting the activities and population sizes of their hosts, which may positively or negatively interact with other community members. Therefore, the actual contributions of different lineages of CPR bacteria to ecological functions in each ecosystem should be studied;

(4) The potential effects on human health—Current studies have already suggested the potential importance of CPR bacteria to human health, mainly because CPR bacteria were detected in different human habitats and their abundance was affected by several pathologies [50]. Although some studies suggest that CPR bacteria may provide antibiotic protection to bacterial host [51,52], the mechanisms related to how CPR bacteria affect human health still remain elusive. Thus, it is necessary to systematically investigate the species diversity and physiological and genomic features of CPR bacteria, as well as the interactions and coevolutions between CPR bacteria and their hosts in human body habitats, such as the oral cavity and intestine, which consequently affect human health.

## Figures and Tables

**Figure 1 microorganisms-10-02232-f001:**
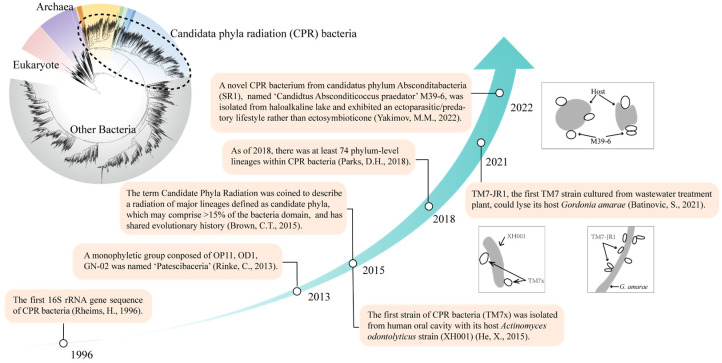
Research history of candidate phyla radiation (CPR) bacteria [1,2,8,9,12,13,14].

**Figure 2 microorganisms-10-02232-f002:**
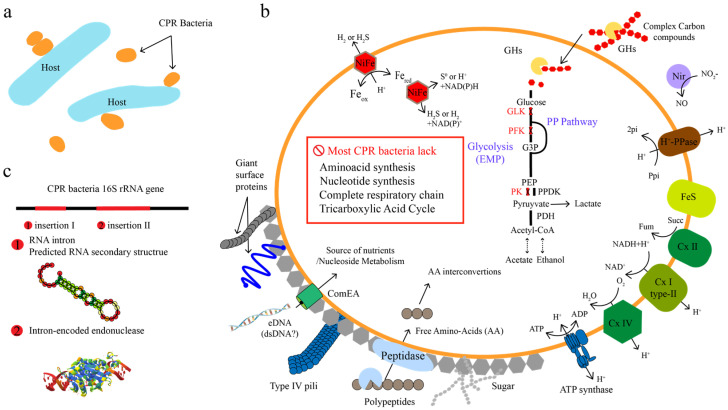
The most common features of CPR bacteria. (**a**) Symbiotic lifestyle of CPR bacteria. (**b**) Metabolic and cell features of CPR bacteria. (**c**) Intron-encoding 16S rRNA gene of CPR bacteria. Abbreviations not defined in the text: GLK, glucokinase; PFK, phosphofructokinase; PK, pyruvate kinase; PPDK, pyruvate phosphate dikinase; Ppi, pyrophosphate; Pi, inorganic phosphate; Cx I, complex I (NADH dehydrogenase); Cx II, complex II (succinate dehydrogenase); Cx IV, cytochrome c oxidase. Protein Data Bank structure was used as the template for structural modeling (c②)—PDB ID: 1R7M [22].

**Table 1 microorganisms-10-02232-t001:** Summary of parasitic/symbiotic interrelationships between CPR bacteria and their hosts.

Phylum	Organisms	Host Organism	Isolate Source	Reference
Saccharibacteria (TM7)	TM7x	*Actinomyces odontolyticus* strain XH001	Human oral cavity	[12]
	TM7-JR1	*Gordonia amarae*	Wastewater treatment	[13]
Absconditabacteria (SR1)	*Candidatus Vampirococcus lugosii* *	Halochromatium-like γ-proteobacterium	Hypersaline lake	[19]
	*Ca. Absconditicoccus praedator* M39-6	Halorhodospira halophila	Haloalkaline Lake	[14]
Yanofskybacteria	Ca. Yanofskybacteria PMX_810_sub	*Methanothrix*	Wastewater treatment	[15]
Parcubacteria	*Candidatus Sonnebornia yantaiensis* *	*Chlorella*/*Paramecium bursaria*	Freshwater pond	[27]

* indicates that the strain was not isolated in pure culture.

## Data Availability

Not applicable.

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
