# Peer review of "Widespread but Poorly Understood Bacteria: Candidate Phyla Radiation"

_microorganisms, 2022, doi:10.3390/microorganisms10112232_

Round 1

Reviewer 1 Report

The work ¨Widespread but poorly understood bacteria: Candidate Phyla Radiation” represents an important review of data in the field. It describes and summarize CPR bacteria by covering relevant information from different sources involving several areas in which this bacterial division has been reported.

I was reviewing the state of art for CPR and found no similar document with this approach; so, in this line, this represents an opportunity for authors and for the Journal too.

Each section in the document has been conveniently added, in a sort of novel grouping of data with no obvious or previous approach. In this regard, it is important to recognize that an important amount of information is given across paragraphs which can be highly valuable; however, this condition also means that authors should be able of giving to the reader as much “key spots” as possible, to make this effort of reviewing this data fruitful. Although sometimes “Tables” are annoying and I´m not a Tables´ fan, my perception on this document is that an important accumulative information should be summarized in a Table for this review. For instance, consistent with the “Perspective” section, there is an important element considering species/host(s), which should be useful for readers to be found in a ready-to-go kind (Table). May be a Table including “a few” relevant aspects considered in the “Perspective” section could be a quite good final effort (an inductive summary) for the document, with no relevant publishing delay.

It is my feeling that this review can be published with no additional effort on it, however, my point is that once authors are on the road, why not this additional effort is given in the paper.

Both Figures in the document are quite illustrative, leading to the reader to acquire a very clear idea about CPR.

Author Response

The work ¨Widespread but poorly understood bacteria: Candidate Phyla Radiation” represents an important review of data in the field. It describes and summarize CPR bacteria by covering relevant information from different sources involving several areas in which this bacterial division has been reported.

I was reviewing the state of art for CPR and found no similar document with this approach; so, in this line, this represents an opportunity for authors and for the Journal too.

Each section in the document has been conveniently added, in a sort of novel grouping of data with no obvious or previous approach. In this regard, it is important to recognize that an important amount of information is given across paragraphs which can be highly valuable; however, this condition also means that authors should be able of giving to the reader as much “key spots” as possible, to make this effort of reviewing this data fruitful. Although sometimes “Tables” are annoying and I´m not a Tables´ fan, my perception on this document is that an important accumulative information should be summarized in a Table for this review. For instance, consistent with the “Perspective” section, there is an important element considering species/host(s), which should be useful for readers to be found in a ready-to-go kind (Table). May be a Table including “a few” relevant aspects considered in the “Perspective” section could be a quite good final effort (an inductive summary) for the document, with no relevant publishing delay.

It is my feeling that this review can be published with no additional effort on it, however, my point is that once authors are on the road, why not this additional effort is given in the paper.

Both Figures in the document are quite illustrative, leading to the reader to acquire a very clear idea about CPR.

Reply: Thanks, a table containing the key spots of this review has been added in our revised manuscript.

Reviewer 2 Report

The review entitled "Widespread but poorly understood bacteria: Candidate Phyla Radiation" is well written and reports clearly and in-depth way the knowledge acquired so far on. I only suggest to improving the quality of all figures and fix some typo and incorrect formatting. (L81, L125, 186).

Author Response

The review entitled "Widespread but poorly understood bacteria: Candidate Phyla Radiation" is well written and reports clearly and in-depth way the knowledge acquired so far on. I only suggest to improving the quality of all figures and fix some typo and incorrect formatting. (L81, L125, 186).

Reply: Thank you very much for this very positive comment, the manuscript has been improved according to the suggestions!

Reviewer 3 Report

Authors present a comprehensive review on the widespread of bacteria belonging to the division of Candidate Phyla Radiation. The summarized data are useful for a reader to become more acquainted with this group of exciting bacteria. In addition, authors highlighted challenges and further research trends related to a deeper understanding of physiology including pathology, biochemistry and ecology of Candidate Phyla Radiation.

Minor issues

Lines 99-102. Consequences of a lack of some ribosomal proteins (as well as truncation of some rRNAs) on ribosome features/functioning could be more deeply discussed.

A recent example of symbiosis of the representative of Candidate Phyla Radiation should be addressed (Kuroda et al. Symbiosis between Candidatus Patescibacteria and Archaea Discovered in Wastewater-Treating Bioreactors. mBio, 2022 13(5):e0171122. doi: 10.1128/mbio.01711-22).

More deep discussion on impact of Candidate Phyla Radiation on health and disease as well as appropriate citations for interested readers should be included (for example, Naud et al. Candidate Phyla Radiation, an Underappreciated Division of the Human Microbiome, and Its Impact on Health and Disease. Clin. Microbiol. Rev. 2022, e0014021. doi: 10.1128/cmr.00140-21; Maatouk et al. New Beta-lactamases in Candidate Phyla Radiation: Owning Pleiotropic Enzymes Is a Smart Paradigm for Microorganisms with a Reduced Genome. Int. J. Mol. Sci. 2022, 23, 5446. https://doi.org/10.3390/ijms23105446).

Author Response

Authors present a comprehensive review on the widespread of bacteria belonging to the division of Candidate Phyla Radiation. The summarized data are useful for a reader to become more acquainted with this group of exciting bacteria. In addition, authors highlighted challenges and further research trends related to a deeper understanding of physiology including pathology, biochemistry and ecology of Candidate Phyla Radiation.

Minor issues

Lines 99-102. Consequences of a lack of some ribosomal proteins (as well as truncation of some rRNAs) on ribosome features/functioning could be more deeply discussed.

Reply: Thanks for this helpful comment. Although, there is now no study showing how the lack of some ribosomal proteins in CPR bacteria affect the ribosome function. But, some studies demonstrated that due to the missing some regions in approximately half of the ribosomal proteins as well as in 16S, 23S, 5S rRNAs in CPR bacteria, ribosomes were predicated to be smaller in CPR bacteria than those in free-living non-CPR bacteria. This information has been added in the revised manuscript.A recent example of symbiosis of the representative of Candidate Phyla Radiation should be addressed (Kuroda et al. Symbiosis between Candidatus Patescibacteria and Archaea Discovered in Wastewater-Treating Bioreactors. mBio, 2022 13(5):e0171122. doi: 10.1128/mbio.01711-22).Reply: Thanks for this meaningful comment. It’s our mistake to miss such an important example, and it has been summarized and added in our revised manuscript.

More deep discussion on impact of Candidate Phyla Radiation on health and disease as well as appropriate citations for interested readers should be included (for example, Naud et al. Candidate Phyla Radiation, an Underappreciated Division of the Human Microbiome, and Its Impact on Health and Disease. Clin. Microbiol. Rev. 2022, e0014021. doi: 10.1128/cmr.00140-21; Maatouk et al. New Beta-lactamases in Candidate Phyla Radiation: Owning Pleiotropic Enzymes Is a Smart Paradigm for Microorganisms with a Reduced Genome. Int. J. Mol. Sci. 2022, 23, 5446. https://doi.org/10.3390/ijms23105446).

Reply: Thanks for this important suggestion, the information has been summarized and added in the new version!
